# Reactions in the Radiosensitizer Misonidazole Induced by Low-Energy (0–10 eV) Electrons

**DOI:** 10.3390/ijms20143496

**Published:** 2019-07-16

**Authors:** Rebecca Meißner, Linda Feketeová, Eugen Illenberger, Stephan Denifl

**Affiliations:** 1Institut für Ionenphysik und Angewandte Physik and Center for Molecular Biosciences Innsbruck, Leopold-Franzens Universität Innsbruck, Technikerstrasse 25, A-6020 Innsbruck, Austria; 2Atomic and Molecular Collisions Laboratory, CEFITEC, Department of Physics, Universidade NOVA de Lisboa, 2829-516 Caparica, Portugal; 3Université de Lyon; Université Claude Bernard Lyon1; Institut de Physique Nucléaire de Lyon, CNRS/IN2P3 UMR 5822, 69622 Villeurbanne CEDEX, France; 4Institut für Chemie und Biochemie-Physikalische und Theoretische Chemie, Freie Universität Berlin, Takustrasse 3, 14195 Berlin, Germany

**Keywords:** electron attachment, misonidazole, radiosensitizer, mass spectrometry, fragmentation, nitroimidazoles, reduction

## Abstract

Misonidazole (MISO) was considered as radiosensitizer for the treatment of hypoxic tumors. A prerequisite for entering a hypoxic cell is reduction of the drug, which may occur in the early physical-chemical stage of radiation damage. Here we study electron attachment to MISO and find that it very effectively captures low energy electrons to form the non-decomposed molecular anion. This associative attachment (AA) process is exclusively operative within a very narrow resonance right at threshold (zero electron energy). In addition, a variety of negatively charged fragments are observed in the electron energy range 0–10 eV arising from dissociative electron attachment (DEA) processes. The observed DEA reactions include single bond cleavages (formation of NO_2_^−^), multiple bond cleavages (excision of CN^−^) as well as complex reactions associated with rearrangement in the transitory anion and formation of new molecules (loss of a neutral H_2_O unit). While any of these AA and DEA processes represent a reduction of the MISO molecule, the radicals formed in the course of the DEA reactions may play an important role in the action of MISO as radiosensitizer inside the hypoxic cell. The present results may thus reveal details of the molecular description of the action of MISO in hypoxic cells.

## 1. Introduction

A variety of nitroimidazole derivatives have been under investigation for their potential use in cancer therapy [1,2]. Here we study free electron attachment to the gas phase nitroimidazolic derivative misonidazole (MISO) (C_7_H_11_N_3_O_4_; see Figure 1a for the molecular structure) which was clinically tested in trails as radiosensitizer for the treatment of hypoxic tumors [2]. Such tumors are characterized by a significant low oxygen state compared to the normal cell tissue. As radiosensitizer, MISO should be preferentially cytotoxic to hypoxic cells. The molecular mechanisms how radiosensitizers like MISO operate, have not been proven yet. Previously it was suggested that this cellular effect is only produced after reduction of the drug [3]. In cancer therapy, reduction of MISO is performed with zinc, ammonium, or by radiolysis by high-energy quanta (particles or photons in the MeV range). The subsequent action of radiosensitizers should ideally result in a selective death of the tumor cells without damage of healthy tissue [4,5,6]. In particular, within the complex processes which finally lead to cell damage, reactions induced by low-energy electrons with kinetic energies between ~0 and 10 eV may play an important role in the early time window (<ps) after energy deposition [4,5,6,7]. This hypothesis comes from the fact that in the first step the action of high-energy quanta (photons or particles) with living cells removes electrons from the molecular network of the cell by various ionization mechanisms [8]. These ballistic secondary electrons are quickly slowed down and can initiate the reduction of a radiosensitizer as well as effective chemical reactions before they enter some stage of solvation and become a chemically inactive species. The estimated quantity is 10^4^–10^5^ secondary electrons per 1 MeV primary quantum [9]. In the very low energy domain (0–3 eV) and before reaching some stage of solvation, these ballistic secondary electrons can initiate chemical reactions via dissociative electron attachment (DEA) which lead to the formation of negatively charged fragment ions and radicals [10,11,12,13,14]. Due to their potentially high cross sections, it was believed that DEA reactions occurring at low energies (0–3 eV) within the ps time window after energy deposition are significant and decisive steps towards cell apoptosis [4,5,6].

Radiation damage can involve any of the cell components like DNA, water, and/or proteins. Ionization of water leads to the formation of highly reactive OH radicals which may attack important cell components. It is estimated that damage of the genome in a living cell by high energy radiation is about one third direct and two third indirect [15]. Direct damage is related to the energy deposition directly in the DNA and closely bound water molecules. In contrast, indirect damage is caused by energy deposition in the nearer vicinity of DNA. Since this mainly involves ionization of water, indirect damage is mainly ascribed to the action of the above-mentioned OH radicals [16].

Previous studies showed that DEA can be an effective process in breaking bonds and hence generates radicals at electron energies considerably below typical bond dissociation energies [10,11,12,13,14]. Such behavior was also observed in DEA studies with radiosensitizers like modified nucleobases [5,6,17,18,19,20]. By modification of the native nucleobase by highly electron affinic side groups, an increase of the DEA cross section can be achieved. It is thus likely that in the course of radiotherapy treatment, radicals from the radiosensitizer damage the tumor cells. Therefore, for the characterization of a radiosensitizer it is of particular significance to know its response towards low-energy electrons.

As member of the class of 2-nitroimidazoles (the NO_2_ group binds to the imidazole ring at the C2 position), MISO was previously tested in clinical trials as radiosensitizer for hypoxic tumors. Though considered to be a highly efficient radiosensitizer, the trials turned out to be unsuccessful due to the high neurotoxicity of MISO [21]. This side effect does not allow the application of the drug in the required doses. In order to gain knowledge on mechanisms of radiosensitizers on the molecular level upon irradiation, we have studied electron attachment to the MISO compound. As will be shown below, MISO is in fact very effectively reduced by capturing low energy electrons. This proceeds via (a) formation of the intact molecular anion at very low energies (close to 0 eV) and (b) in the range 0–10 eV, by generating NO_2_^−^ and a variety of further fragment anions via DEA thus revealing in detail the possible reduction processes. In addition, the radicals generated in the course of the DEA reactions are presumably relevant in the action of MISO as radiosensitizer inside the hypoxic cell.

## 2. Results and Discussion

### 2.1. General Features of Associative (AA) and Dissociative Electron Attachment (DEA) and Characterization of the Involved Resonances

The dominant process in electron attachment to MISO is the very effective formation of the non-decomposed parent anion which is exclusively formed within a very narrow resonance close to 0 eV (Figure 2). We further observe a large variety of negatively charged fragment ions generated within pronounced resonances, which are the result of dissociative electron attachment (DEA, Figure 3, Figure 4 and Figure 5), among them the prominent DEA reaction yielding NO_2_^−^ representing a simple (C–NO_2_) bond cleavage (Figure 3a). Further DEA reactions involve multiple bond cleavages (simultaneous loss of a neutral NO_2_ and a CH_2_ unit) (Figure 3c), complex reactions associated with rearrangement in the transitory negative ion (TNI), followed by multiple bond cleavages and formation of new molecules (loss of a neutral H_2_O molecule) (Figure 4a), excision of a CN^−^ and OCN^−^ unit (Figure 4b,c) and formation of the dehydrogenated nitroimidazole anion (Figure 5a). All these DEA reactions are observed in the electron energy range 0–10 eV and at significant lower cross sections compared to the associative attachment process generating the parent anion.

The formation of a non-decomposed parent anion by capture of a free electron in the gas phase under collision free conditions requires that the excess energy deposited by the attachment (comprised of the initial kinetic energy of the incoming electron and the electron affinity of the molecule) is effectively dispersed over the vibrational degrees of freedom in the TNI. In such case, autodetachment is delayed and the resulting lifetimes are in the µs regime and longer, which allow observation by mass spectrometry [13]. MISO has a positive electron affinity of 1.33 eV [22]. The Singly Occupied Molecular Orbital (SOMO) of the MISO anion is shown in Figure 1b and clearly indicates the delocalization of the excess electron over the whole nitroimidazole group. In contrast, resonant (dissociative) electron attachment beyond 0 eV is usually described as accommodation of the extra electron into one of the normally unoccupied molecular (valence) orbitals (MOs) thus forming the temporary negative ion (TNI) or synonymously, the resonance which then decomposes. In terms of localized Born–Oppenheimer (BO) potential energy surfaces, DEA is then described as a vertical transition between the potential energy surface of the neutral and that of the anion followed by dissociation into a stable fragment anion and the neutral counterpart [10,13]. It should be emphasized that AA close to 0 eV is a non-BO phenomenon which cannot be described as a transition between localized potential energy surfaces.

#### 2.1.1. Formation of the Non-Decomposed Parent Anion

As mentioned above, the intact molecular anion of MISO is the most abundant anion observed in the present experiment. Though we did not explicitly determine the absolute cross section for this AA reaction, we estimate from the relative ion yields recorded at the corresponding partial pressures that the cross section has a similar order of magnitude as the well-known electron scavengers like SF_6_ (formation of SF_6_^−^) [23] and CCl_4_ (Cl^−^ formation) [24]. SF_6_ has one of the highest electron attachment cross sections known [23]. One of the main reasons for this behavior is the fact that all DEA channels are endothermic and therefore not available at electron energies close to 0 eV. Together with the high symmetry of the molecule, the conditions for the formation of a metastable parent anion are fulfilled, where spontaneous autodetachment is the only competitive channel. Additionally, many other examples of molecules creating a metastable parent anion have been already reported. Their ion signal is usually characterized by an exclusive narrow peak at the electron energy of ~0 eV [13,25,26] or in combination with another peak at slightly higher electron energies [27,28,29]. The only remarkable exception is the C_60_^−^ ion yield formed upon electron attachment to C_60_ [30,31]. This anion is formed over a wide range of electron energies extending up to about 12 eV. C_60_ is of high symmetry and the binding energies for each C atom are equal. This provides ideal conditions for delayed autodetachment and the absence of DEA [30,31].

In contrast, MISO is of low symmetry but the appreciable number of 69 vibrational degrees of freedom apparently provides an effective means for energy redistribution making the observation of an intense parent anion possible. We note that MISO exhibits a similar behavior towards attachment of a single electron like the 5-nitroimidazolic molecule nimorazole. The latter compound is utilized as radiosensitizer for the treatment of pharyngeal and supra-glottic carcinoma in Danish radiotherapy centers [1]. A recent electron attachment study showed that the parent anion is the most abundant anion for nimorazole, and DEA plays a minor role [32]. Due to its morpholine ring linked by a short hydrocarbon chain to the nitroimidazole moiety, even more excess energy could be stored than in the case of MISO. Therefore, one may be tempted to conclude that just a large number of vibrational degrees is required to cause sufficient stabilization of the transient negative anion. However, in that context we also mention that in a recent study by our laboratory with the considerable smaller methylated nitroimidazoles 1-methyl-4-nitroimidazole and 1-methyl-5-nitroimidazole (36 vibrational degrees of freedom) the non decomposed molecular anion was also observed in both compounds appearing at a narrow resonance right at threshold [33,34]. Since for the non-methylated nitroimidazoles no parent anion is observable on µs-timescales, the replacement of the hydrogen at the N1 position of the imidazole ring by the methyl group closes DEA channels at threshold and allows the stabilization of the molecular anion to mass spectrometric time scales.

#### 2.1.2. Dissociative Electron Attachment (DEA)

The ion yields due to the different dissociative attachment reactions appear, depending on the ion under observation, within resonances extending from threshold (0 eV) to about 10 eV (Figure 3, Figure 4 and Figure 5). The evolution of these resonances finally results in simple bond cleavages (formation of NO_2_^−^) but also much more complex reactions in the TNI finally leading to the loss of a neutral H_2_O unit, excision of CN^−^, etc.

In cases when DEA is operative already at threshold, we have to assume that vibrational Feshbach resonances (VFRs) are involved. This is in particular the case in the DEA reaction leading to NO_2_^−^ which shows two overlapping narrow peaks at very low electron energies (vide infra).

##### Formation of NO_2_^−^, Loss of the Neutral Unit CH_3_, and Loss of the Two Neutral Units NO_2_ + CH_2_

Figure 3 presents the ion yields of the corresponding DEA reactions. NO_2_^−^ (Figure 3a) is the dominant DEA product representing the cleavage of a C–NO_2_ bond
e^−^ + MISO → MISO^#–^ → (MISO – NO_2_) + NO_2_^−^,(1)
with MISO^#–^ as the transitory negative ion of MISO formed upon electron attachment.

The NO_2_^−^ fragment anion is formed via two narrow and overlapping features close to threshold, a further resonance centered around 1.5 eV and a broad and unstructured feature in the energy range between 2.5 and 4.5 eV. We assign the resonances off 0 eV as shape resonances (with possible contributions of low-lying core excited resonances in the broad feature peaking at ~3 eV [35]) and the narrow features close to 0 eV as vibrational Feshbach resonances (VFRs) in analogy to the situation previously described in DEA to nitroimidazoles [33,34]. Such VFRs can in fact couple to dissociative valence configurations, thereby acting as effective doorways for DEA [36].

Our quantum chemical calculations on the thermodynamics of NO_2_^−^ formation upon DEA to MISO indeed indicate a more complex DEA mechanism. At first, we calculated the free reaction energy for the DEA Reaction (1), where a simple bond cleavage reaction is assumed. In this case, the reaction is endothermic with a free energy of +0.339 eV. Since the ion yield of NO_2_^−^ is observed already at threshold, such simple bond cleavage reaction does not lead to the observed threshold peak. Therefore, we computationally investigated rearrangement reactions and found that NO_2_ loss with H transfer to C2 position of the imidazole moiety (see Figure 6) gives an exothermic reaction with a free energy of −0.341 eV.

For the NO_2_^−^ peaks at the electron energy of about 1.5 and 3 eV a single bond cleavage is energetically possible. We note that the ion yield is similar to the NO_2_^−^ ion yield formed upon DEA to nitroimidazole [33]. Kossoski and Varella performed theoretical calculations of low-energy resonances in 4- and 5-nitroimidazole (NI) and 1-methyl-nitroimidazoles [37]. They suggest an indirect dissociation mechanism for the NO_2_^−^ fragment anion, where coupling of π* states and the repulsive σ*_CN_ state occurs. By the analogy of the ion yield we may therefore assume also an indirect dissociation mechanism for MISO, which involves the coupling of the π* state and the repulsive σ*_CN_ state.

Figure 3b shows the fragment anion which is formed by the loss of a neutral CH_3_ unit according to
e^−^ + MISO → MISO^#–^ → (MISO – CH_3_)^−^ + CH_3_,(2)
and hence the cleavage of the O–CH_3_ bond with the excess charge finally localized on the large imidazole containing unit. This negatively charged fragment is only observed within the two very narrow features close to threshold assigned as VFRs, indicating that the evolution of the shape resonances leading to NO_2_^−^ does not result in Reaction (2).

This observation mirrors the possibility to distribute the excess energy which is different for the two DEA Reactions (1) and (2). The excess energy in a DEA reaction amounts to the electron energy above the thermodynamic threshold energy of the respective process and is distributed among the formed fragments.

In Reaction (1), the light fragment ion NO_2_^−^ is detected while the excess energy in the large neutral fragment (M–NO_2_) may lead to further and even multiple decompositions. In contrast, in Reaction (2) from stoichiometry we know that the neutral fragment is CH_3_ (provided that further decompositions can be excluded) but for the large ionic fragment the ability to carry excess energy is limited by the decomposition threshold with respect to both, detachment of the extra charge and dissociation. This fact apparently restricts the observation of the large ionic fragment to the very low energy domain.

Figure 3c finally shows the yield for the ionic fragment with mass 141 u, formed in the DEA reaction formally associated with the loss of CH_2_ + NO_2_.
e^−^ + MISO → MISO^#–^ → (MISO – CH_2_ – NO_2_)^−^ + CH_2_NO_2_.(3)

The corresponding ion yield shows only one asymmetric resonance feature close to threshold, which is in contrast to Reactions (1) and (2), where the related ion yields close to threshold are characterized by a structure with two distinct peaks. We note that the loss of a methyl group and the NO_2_ would only require two simple bond cleavages. However, the loss of CH_2_ as observed in the present experiment is only possible by a rearrangement reaction. We computationally investigated various rearrangement reactions and found as lowest possible free reaction energy a value of +0.11 eV, which involves migration of the hydrogen to the C2 carbon site and formation of neutral CH_2_NO_2_ (see Figure 6). Experimentally, a peak maximum at 0.05 ± 0.01 eV is obtained (the stated error corresponds to the step width of the electron energy scan), which is slightly below the predicted onset. Therefore, this ion yield may be interpreted as hot band transition which can play a significant role in DEA in particular at elevated temperatures as is the case in the present experiment. The considerable intensity of such hot band transitions is due to the peculiarities of DEA like increasing cross section with decreasing electron energy, etc. [38,39].

##### Loss of Neutral H_2_O and Excision of the Pseudohalogens CN^−^ and OCN^−^

The ion yields due to the rather complex DEA reactions resulting in the loss of a neutral water unit and the excision of the ions CN^−^ and OCN^−^ are presented in Figure 4. Formation of a neutral water unit following electron attachment to the target compound (Figure 4a) may occur at different sites of the target compound and proposing a detailed reaction mechanism would be rather speculative. A likely site is the linear chain outside the imidazole unit at the [-H_2_C-(CH-OH)-CH_2_-] unit. In this case, the reaction would require the cleavage of a C–H and a C–OH bond followed by the formation of the H_2_O molecule. As is obvious from Figure 4a, this reaction is already operative at threshold (zero electron energy) and additionally within the weak resonances at around 0.25 and 1 eV. We note that the loss of a water molecule upon DEA turned out to be an isomer selective process for NIs [40]. The DEA reaction with formation of H_2_O was only abundant for the 2-NI isomer, while for the 4-NI molecule this channel was very weak. Indeed, a pathway for H_2_O loss was found during the computational exploration of the relevant potential energy surfaces for the 2-NI isomer. The found reaction was exothermic in agreement with the experimental data. The present results also indicate an exothermic reaction for the loss of water. However, the abundance of (MISO – H_2_O)^−^ is rather minor, which may be explained by the efficient stabilization of the MISO parent anion in competition to DEA reaction. In contrast, for 2-NI no parent anion was observable within the detection limit of the experimental apparatus.

Figure 4b represents the ion yield due to the excision of CN^−^ and Figure 4c that of OCN^−^ formation. Both CN and OCN are well known pseudohalogens having electron affinities (EAs) exceeding even those of the halogen atoms (EA (CN) = 3.86 eV, EA (OCN) = 3.61 eV [41]). On the other hand, a large EA does not necessarily lead to a high cross section for the formation of CN^−^ via DEA. For example, CN^−^ formation from compounds like (amino)acetonitrile or benzonitrile [42,43,44] was comparatively weak due to the underlying decomposition mechanism. Although it is also formed for these compounds via a single C–CN bond cleavage, the dissociation mechanism was suggested to be indirect in the way that the excess electron initially resides in a π* (CN) antibonding MO and decomposition into CN^−^ requires transfer of the available energy from CN into the C–CN coordinate (vibrational predissociation).

Both pseudohalogenide ions appear within resonance features above 2 eV. It is likely that the C–NO_2_ site is involved in the corresponding DEA reaction. While there is no established thermochemical data available for the present system it should be mentioned that DEA to MISO leading to CN^−^ can be accompanied by a more or less complete degradation of the entire target molecule as, besides CN^−^, stable neutral counterparts like N_2_, CO_2_ and hydrocarbons can be formed. It has in fact been demonstrated in DEA to the comparably smaller system acetamide, that excision of CN^−^ observed within a resonance at 2 eV is accompanied by a complete degradation of the entire target molecule [45]. Similar complex decomposition processes are likely accompanied with the formation of OCN^−^, as shown previously for the pyrimidine nucleobases [46].

##### Formation of Nitroimidazolic Anions 

Figure 5 finally presents ion yields of three DEA reactions associated with the cleavage of the N–C bond and hence formation of nitroimidazolic anions (C_3_H_2_N_3_O_2_)^−^, (C_3_H_3_N_3_O)^−^ and (C_3_H_2_N_2_O)^−^. More precisely, Figure 5a shows the ion yield recorded at 112 u which we assign to the dehydrogenated closed shell anion of NI ((C_3_H_2_N_3_O_2_)^−^), a prominent DEA product from the NIs previously studied in our laboratory [33,34]. The ion yield recorded at 96 u, Figure 5b, corresponds to a further loss of a neutral O unit ((C_3_H_3_N_3_O)^−^) and that recorded at 82 u, Figure 5c, to a loss of a neutral NO unit from the (dehydrogenated) NI anion ((C_3_H_2_N_2_O)^−^). While this could lead to anions with an imidazole structure, we cannot exclude that in the course of these DEA reactions the cyclic structure deteriorates.

## 3. Materials and Methods 

### 3.1. Experiment

The electron attachment experiments were performed with a crossed-beam experiment recently described in [32]. The setup comprises a molecular beam source consisting of an oven with capillary with 1 mm inner diameter, a hemispherical electron monochromator (HEM), a quadrupole mass analyzer, and a channel electron multiplier with pulse counting system. Since only gas phase studies can be conducted by mass spectrometric means, the crystalline MISO was evaporated in the oven at around 75 °C. The resulting pressure in the vacuum chamber amounts to 5 × 10^−7^ mbar. The effusive beam crosses the electron beam in the interaction region at the end of the HEM. The energy resolution is a compromise over a high electron current and was set to about 100 meV for the current study. The resolution is determined by the full width at half maximum from the well known sharp 0 eV resonance of SF_6_^−^ (AA) and Cl^−^ from CCl_4_ (DEA). Those ion yield curves additionally serve for calibration of the energy scale. The electron current is monitored by a Faraday cup detector placed behind the interaction region to ensure stable conditions. After the anions were formed by electron attachment, they were extracted by a weak electric field between HEM and quadrupole. Subsequently, they were detected and recorded by a preamplifier and detection unit [34].

The misonidazole sample was purchased from Toronto Research Chemicals Canada with a stated purity of 98% and was used as received.

The utilization of the HEM enabled stable electron beam conditions in the measured electron energy range. Therefore, the intensities of the peaks measured for a mass selected anion are comparable. The only exception occurs for peaks at 0 eV electron energy, where the height of the peaks is underestimated due to the experimental limit in the production of electrons with energies approaching 0 eV as well as the finite energy resolution of the electron beam [24]. The ion yields of the mass selected anions shown in Figure 2, Figure 3, Figure 4 and Figure 5 were recorded at identical conditions (same pressure, electron current etc.) and are presented on a relative scale. However, they were not corrected by the mass transmission of the quadrupole mass analyzer and detection efficiency of the channeltron. This leads to an error in the comparison of relative ion intensities, as discussed in [32]. We further note that for the comparison of the MISO^−^ anion yield with the ion yields from SF_6_ and CCl_4_ (see Section 2.1.1.) an additional error in the determination of the corresponding partial pressures in the chamber arises, see [32].

### 3.2. Calculations

Quantum chemical calculations employing the density functional M062x [47,48] were carried out to calculate free energies of reactions, ∆G. The thermodynamic threshold for a DEA reaction, considering the precursor molecule M and a release of a neutral fragment X, can be expressed by ∆G([M – X]^−^) = DE(M–X)–EA(M – X), where DE(M–X) is the bond dissociation energy and EA(M – X) is the electron affinity of the corresponding fragment. The threshold energy for the experimental observation of [M – X]^−^ in electron attachment experiments coincides with ∆G([M – X]^−^) if the fragments are formed with no excess energy. For the MISO we used the lowest structure reported previously [22]. All structures where optimized at the M062x/6-311+G(d,p) level of theory and basis set with the Gaussian-09D01 programme package [49]. Frequencies were calculated in all cases to confirm that the structures are local minima on the potential energy surface and not the transition states.

## 4. Conclusions

Free electron attachment to the radiosensitizer misonidazole (MISO) in the gas phase predominantly creates the non-decomposed anion which is exclusively formed from a very narrow resonance near zero electron energy. In addition, a large variety of fragment anions are observed from resonance features in the energy range from 0 to 10 eV. These DEA reactions involve simple bond cleavages (formation of NO_2_^−^, loss of CH_3_, etc.) and considerable complex reactions (loss of a neutral water unit, excision of the pseudohalogenide ions CN^−^ and OCN^−^). All these electron attachment processes represent initial reduction of the radiosensitizer, which is necessary for its uptake by a hypoxic tumor cell. The present results hence reveal details of the intrinsic reduction process in MISO. In addition, the various neutral radicals formed along the DEA reactions may represent important components in the description of the molecular mechanisms how the radiosensitizer MISO acts within a hypoxic cell. While the present results reveal intrinsic properties of gas phase MISO, the question is on the relevance of the present results for the action of MISO as radiosensitizer in vivo. Extended electron attachment studies to molecules embedded in clusters and in the condensed phase demonstrated [32,50] that in bound molecules, the intrinsic electron attachment properties are preserved. In other words, electron attachment to bound molecules can still be pictured on a molecular site, i.e., attachment to an individual molecule, which is coupled to a particular environment. In light of that, we conclude that the present results can help to reveal details of the molecular mechanisms, how MISO acts as radiosensitizer in hypoxic tumor cells. Future studies with misonidazole in water clusters may show which reactions observed here will sustain in solution, since energy transfer to the water medium will likely modify the dissociation processes.

## Figures and Tables

**Figure 1 ijms-20-03496-f001:**
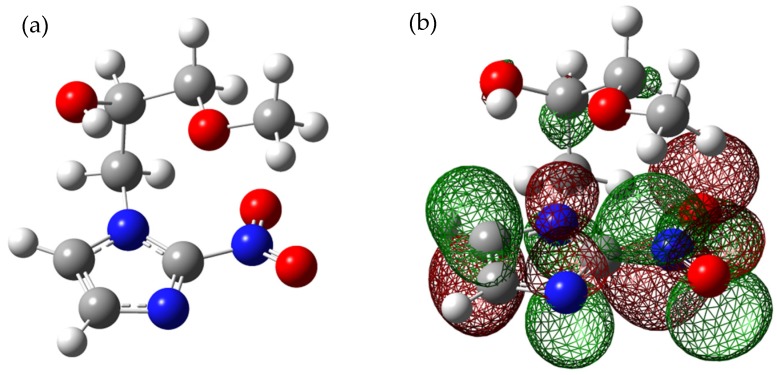
(**a**) Optimized molecular structure of misonidazole (MISO), (**b**) singly occupied molecular orbital of the MISO anion.

**Figure 2 ijms-20-03496-f002:**
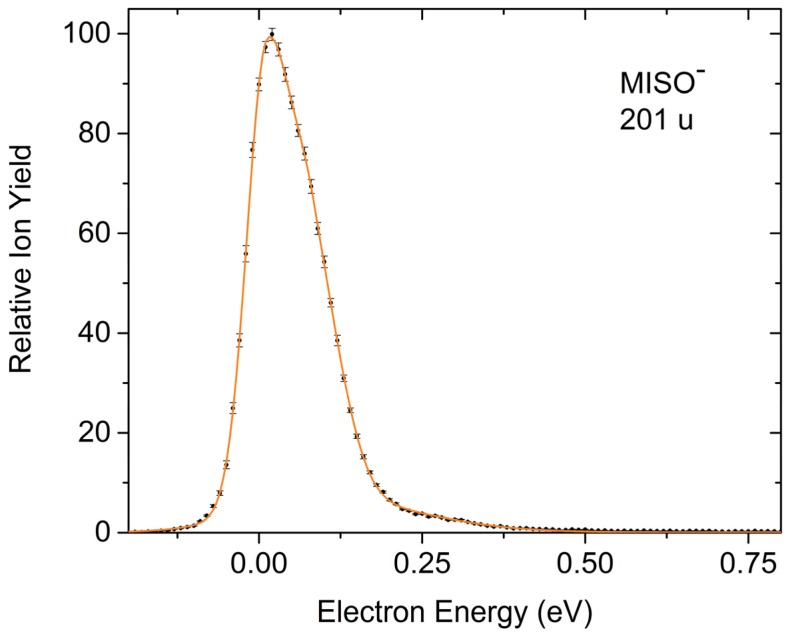
Relative ion yield for the associative attachment (AA) process generating the non-decomposed molecular anion.

**Figure 3 ijms-20-03496-f003:**
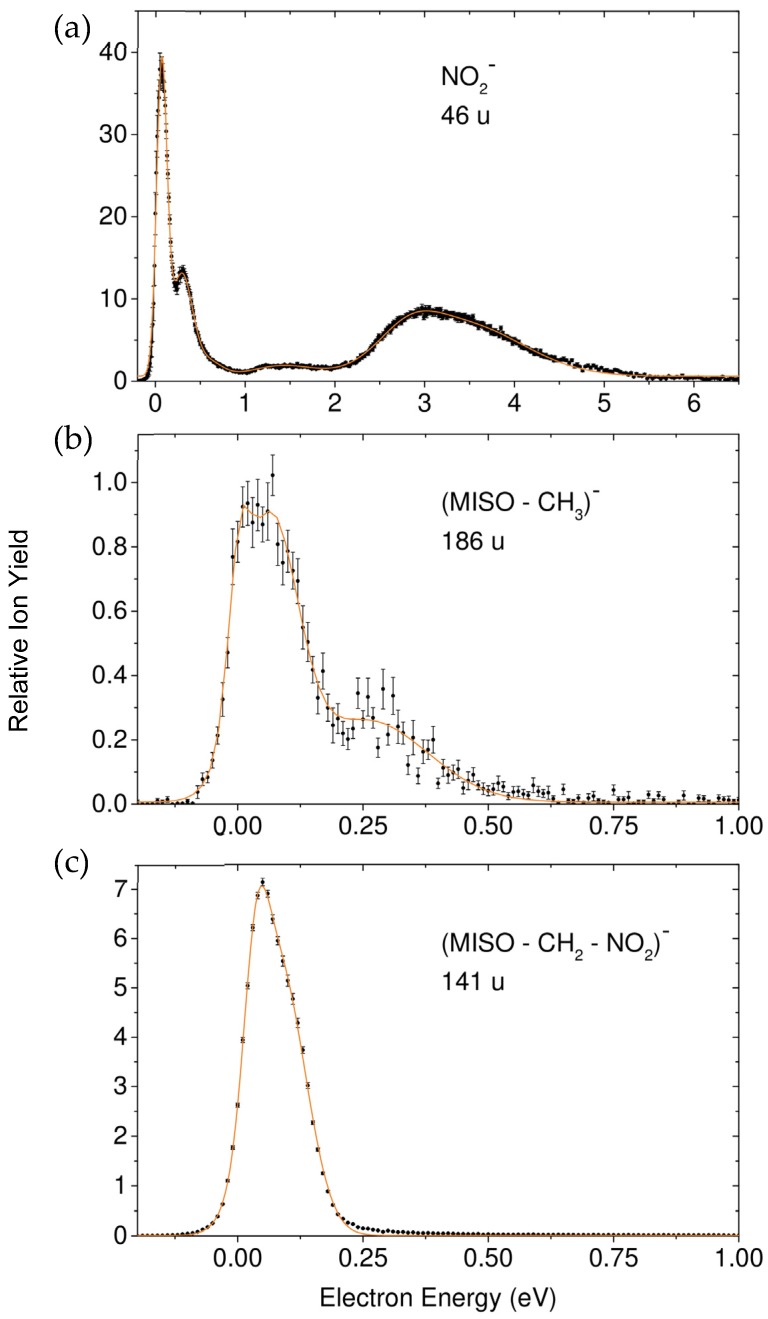
Relative ion yields for the dissociative electron attachment (DEA) reactions generating the NO_2_^−^ fragment ion (**a**), the fragment ion arising from the loss of a neutral CH_3_ unit (M – CH_3_)^−^ (**b**), and the fragment ion due to the loss of the two neutral units CH_2_ and NO_2_ (M – CH_2_ – NO_2_)^−^ (**c**).

**Figure 4 ijms-20-03496-f004:**
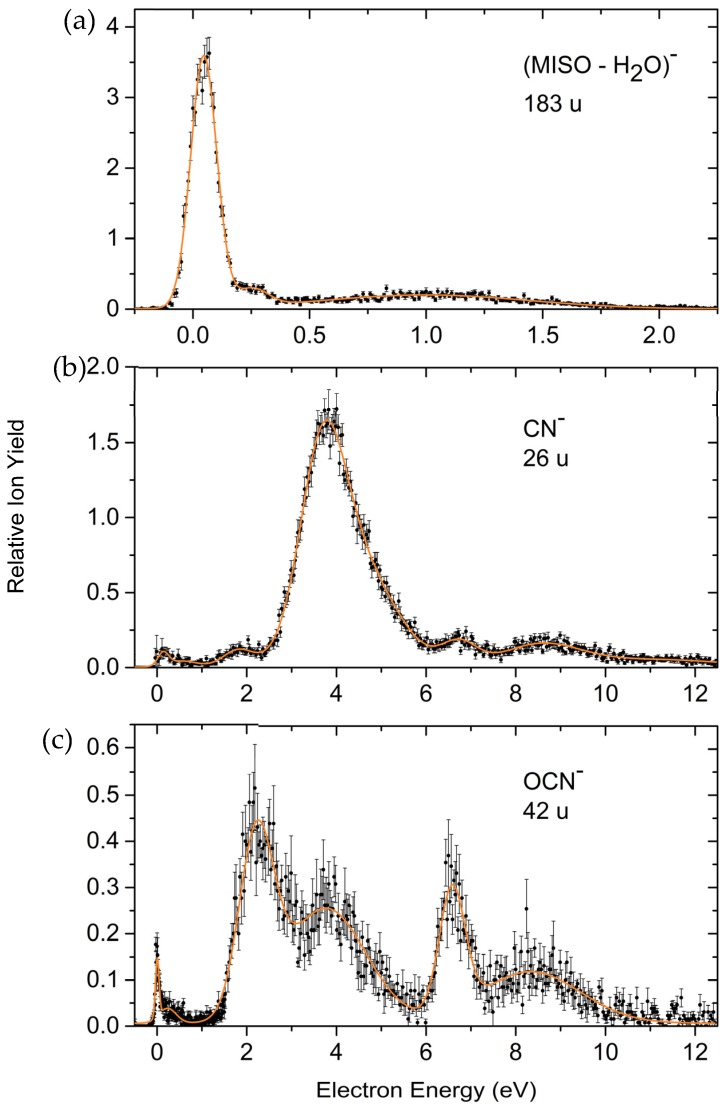
Relative ion yields for the ion appearing from the loss of a neutral water unit (M – H_2_O)^−^ (**a**), relative cross section for the excision of CN^−^ (**b**) and the excision of OCN^−^ (**c**).

**Figure 5 ijms-20-03496-f005:**
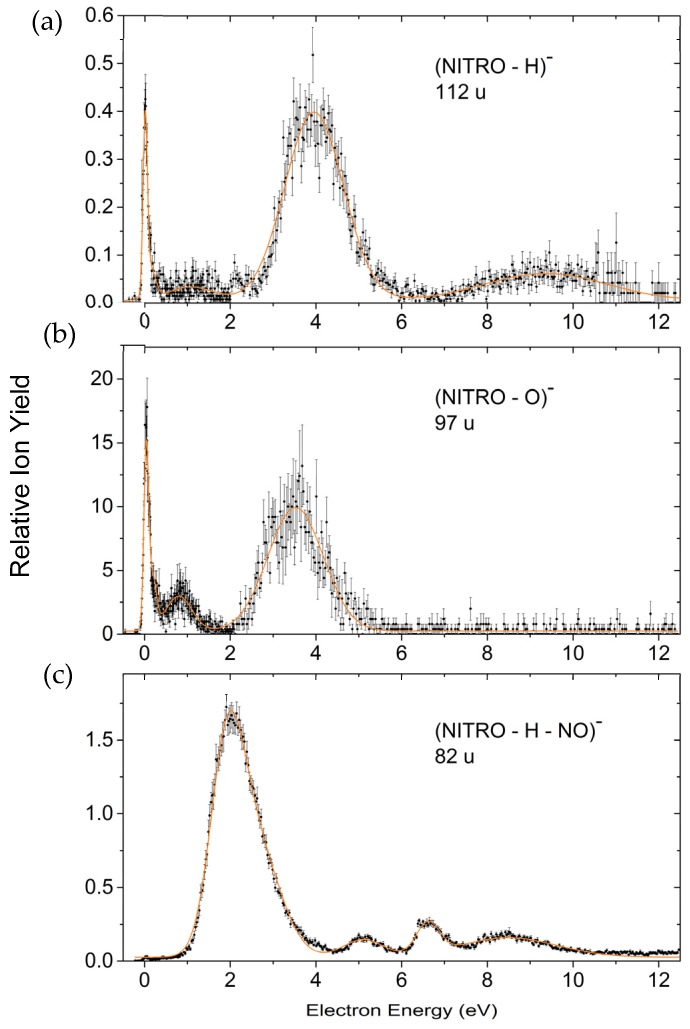
Relative ion yields for the formation of the dehydrogenated closed shell nitroimidazole anion (**a**), the nitroimidazole anion subjected to the additional loss of a neutral O unit (**b**), and the nitroimidazole anion subjected to the additional loss of a neutral NO unit (**c**).

**Figure 6 ijms-20-03496-f006:**
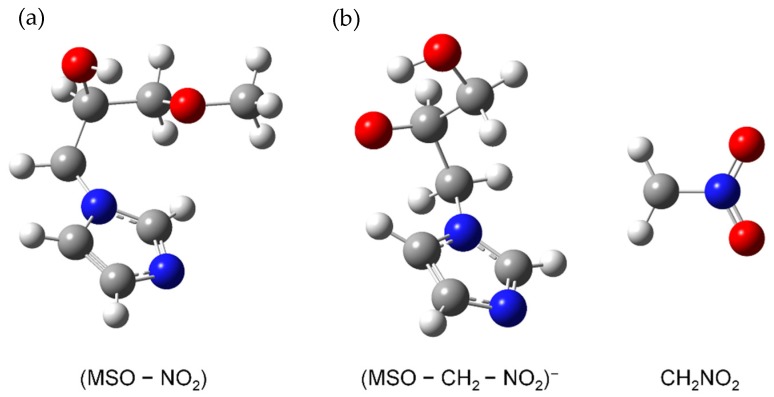
(**a**) Calculated structure of the neutral product of the DEA Reaction (1). (**b**) Calculated structure of the charged and neutral products of the DEA Reaction (3).

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
