# Peer review of "Reactions in the Radiosensitizer Misonidazole Induced by Low-Energy (0–10 eV) Electrons"

_ijms, 2019, doi:10.3390/ijms20143496_

Round 1
Reviewer 1 Report
This well written manuscript describes the products of low energy electron addition to misonidazole which has been used as a radiosensitizer in cancer therapy. The work is done in the gas phase with careful control of electron energy in a crossed beam mass spectrometer. Products are conveniently detected by mass analysis with yields determined as a function of electron energy. All in all an interesting work which gives DEA reactions for electron addition at even zero eV though the major electron reaction at zero eV is the formation of the molecular ion. A few points need revision.
Comments
1. Line 234 “therefore in disagreement with the calculations.” The difference of 0.06 eV is too small to label as in disagreement.
2. 4. Conclusions. The authors suggest the applicability of their results to molecular mechanisms for the radiosensitiser MISO within a hypoxic cell. This is an overstatement since fragmentation reactions are greatly influenced by the first few solvation shell and therefor many routes are suppressed in aqueous solutions owing to energy transfer to the medium. The results for future efforts in clusters and aqueous systems will be more applicable.
Author Response
General reply: We would like to thank the reviewer for his/her report, which helped us to improve the quality of the current work. Our detailed response and the list of corrections addressing all of the comments are presented below, hoping that these are to his/her satisfaction. We also submit the manuscript file, where all changes are highlighted by the word track-correction mode.
This well written manuscript describes the products of low energy electron addition to misonidazole which has been used as a radiosensitizer in cancer therapy. The work is done in the gas phase with careful control of electron energy in a crossed beam mass spectrometer. Products are conveniently detected by mass analysis with yields determined as a function of electron energy. All in all an interesting work which gives DEA reactions for electron addition at even zero eV though the major electron reaction at zero eV is the formation of the molecular ion. A few points need revision.
Comments
1. Line 234 “therefore in disagreement with the calculations.” The difference of 0.06 eV is too small to label as in disagreement.
Answer: We deleted the mentioned statement accordingly.
2. 4. Conclusions. The authors suggest the applicability of their results to molecular mechanisms for the radiosensitiser MISO within a hypoxic cell. This is an overstatement since fragmentation reactions are greatly influenced by the first few solvation shell and therefor many routes are suppressed in aqueous solutions owing to energy transfer to the medium. The results for future efforts in clusters and aqueous systems will be more applicable.
Answer: We agree with the reviewer that the fragmentation pattern observed here for the isolated molecule will not exactly form in the solution phase as well. However, here we investigated possible pathways of fragmentation, where a priori it cannot be ruled out that some of them will occur in solution. In that sense, the present results will help to reveal details on the molecular mechanisms of radiosensitisers and fundamental knowledge on the reduction mechanisms of misonidazole is provided. In addition, we demonstrate that misonidazole is efficiently reduced by low energy electrons, which is important property for its action as radiosensitiser.
Following the comment of the reviewer, we revised the Section 4. Conclusions and pointed out that the fragment pattern will be likely modified by solvation (see manuscript for the changes made).
Reviewer 2 Report
The paper “Reactions in the Radiosensitiser Misonidazole Induced by Low-Energy (0-10 eV) Electrons” by Meisner et al reports an experimental study on low-energy electron attachment to misonidazole molecule, which is considered as a radiosensitizer for the treatment of hypoxic tumors. The experimental results are supported by DFT calculations. The authors report new results, the measurements were carefully performed and the manuscript is clearly and concisely written. The conclusions are sound and the scientific output is of interest for a broad research community. Even though the reported results are obtained from a gas-phase study, they should represent a valuable reference for further “in vivo” research.
I, therefore, recommend the publication of the present manuscript in the IJMS journal.
However, I have several suggestions that authors should consider.
1. Page 4, Line117: “SOMO”?
2. Page 4, Figure 3 (and other relative ion yield curves): are the intensity ratios between different resonant processes relevant? That is, can authors estimate the experimental influence to the shape of the relative ion yields (RIY) curves (such as the effective electron flux, the beam overlap - interaction region, etc). This might be critical when the electron energy approaches zero eV.
3. Can different RIYs intensities (for different fragmentation channels) be compared? The Y-scale reads “relative ion yield”, but the numbers and statistics are clearly different suggesting the intensity of the corresponding process. Therefore, the authors should comment on and extend the experimental section to discuss the detection efficiency (including the efficiency of ion extraction from the interaction region) as a function of m/z.
4. Page 6, Line 136: “Though we did not explicitly determine … ” Although this statement seems sound, for a publication, the authors should include in the Experimental part a more elaborative explanation of how they arrived at this conclusion. For example, are the molecules measured at the same time, how the partial pressures were determined (if by IG, were the corrections considered), did they consider the change of the overlapping region (and its influence) due to the different molecular sizes and possible pressures, etc. This is particularly important since the above statement is one of the most important conclusions of the study.
5. Page 7, Lines 188-195: It is interesting that both negative and positive are of exactly the same absolute value (0.34), although obtained by different calculating procedures. Maybe, authors, cold include the third decimal, to show that this is not a typo.
6. Page 9, Line 233: 0.05+/-0.01: how this error was obtained?
7. Page 9, Line 278: “nitroimidazolic anions”: it would be helpful to give the formula
Page 10, Line 299: “… they were extracted by a weak electric field between …”: see point 3.
Author Response
General reply: We would like to thank the reviewer for his/her report, which helped us to improve the quality of the current work. Our detailed response and the list of corrections addressing all of the comments are presented below, hoping that these are to his/her satisfaction. We also submit the manuscript file, where all changes are highlighted by the word track-correction mode.
The paper “Reactions in the Radiosensitiser Misonidazole Induced by Low-Energy (0-10 eV) Electrons” by Meisner et al reports an experimental study on low-energy electron attachment to misonidazole molecule, which is considered as a radiosensitizer for the treatment of hypoxic tumors. The experimental results are supported by DFT calculations. The authors report new results, the measurements were carefully performed and the manuscript is clearly and concisely written. The conclusions are sound and the scientific output is of interest for a broad research community. Even though the reported results are obtained from a gas-phase study, they should represent a valuable reference for further “in vivo” research.
I, therefore, recommend the publication of the present manuscript in the IJMS journal.
However, I have several suggestions that authors should consider.
1. Page 4, Line117: “SOMO”?
Answer: We defined “SOMO” as Singly Occupied Molecular Orbital in the revised manuscript and added this definition to the list of abbreviations (by replacing HOMO, which is not used).
2. Page 4, Figure 3 (and other relative ion yield curves): are the intensity ratios between different resonant processes relevant? That is, can authors estimate the experimental influence to the shape of the relative ion yields (RIY) curves (such as the effective electron flux, the beam overlap - interaction region, etc). This might be critical when the electron energy approaches zero eV.
Answer: In the present study we utilized an electron monochromator. This device generates low electron currents in the nA regime, which will not lead to considerable space charge effects deteriorating the electron beam profile. The only (well-known) exception, as mentioned by the referee, is the case when electrons have very low velocities. This leads to a limit in the lowest energy one can experimentally achive. If the cross section follows a reciprocal dependence with electron energy, the measured intensity does not reflect the cross section at zero eV. In addition, the height of a zero eV peak (which may have a resonance width below the energy resolution of the beam) also depends then on the electron energy resolution, as described by Hotop and coworkers, Int. J. Mass Spect. 205 (2001) 93-110. Therefore, the intensity ratio of resonances are comparable except at zero eV energy. We mention this point in the Method´s section 3.1.Experiment of the revised manuscript and added the above mentioned reference.
3. Can different RIYs intensities (for different fragmentation channels) be compared? The Y-scale reads “relative ion yield”, but the numbers and statistics are clearly different suggesting the intensity of the corresponding process. Therefore, the authors should comment on and extend the experimental section to discuss the detection efficiency (including the efficiency of ion extraction from the interaction region) as a function of m/z.
Answer: DEA ion yields of various mass selected anions shown would be completely relative, if (i) they are recorded at identical pressures conditions, (ii) identical electron currents, (iii) identical extraction conditions (see the comment of the reviewer below), (iv) uniform mass transmission of the quadrupole mass analyser, and (v) uniform detection efficiency. For the ion yields shown in the figures we can ensure that conditions (i-iii) are fulfilled (for (i)+(ii) see answer above). Regarding (iii) it should be mentioned that the ion extraction fields are chosen such that ions with low and high kinetic energy are approximately discriminated in the same manner, when extracted out of the ion source. The effects of (iv) and (v) are unknown for the present experiment and lead to an error in the estimation of relative intensities.
We therefore mention in the revised manuscript (section 3.1.) that the ion yields were determined at identical conditions of the electron monochromator but they were not corrected regarding mass transmission of the quadrupole analyzer and detection efficiency. For a quantitative estimation of the resulting uncertainty we refer to ref. 32 of the manuscript.
4. Page 6, Line 136: “Though we did not explicitly determine … ” Although this statement seems sound, for a publication, the authors should include in the Experimental part a more elaborative explanation of how they arrived at this conclusion. For example, are the molecules measured at the same time, how the partial pressures were determined (if by IG, were the corrections considered), did they consider the change of the overlapping region (and its influence) due to the different molecular sizes and possible pressures, etc. This is particularly important since the above statement is one of the most important conclusions of the study.
Answer: We refer to our answer to the comment 3, since the same possible uncertainty sources (i)-(v) apply also for the comparison of the MISO anion yield and the ion yields from SF6 and CCl4. We mention in the revised manuscript, that additional uncertainty is given in the determination of the partial pressures and refer to the detailed discussion in ref. [32].
5. Page 7, Lines 188-195: It is interesting that both negative and positive are of exactly the same absolute value (0.34), although obtained by different calculating procedures. Maybe, authors, cold include the third decimal, to show that this is not a typo.
Answer: We added the third decimal, as requested by the reviewer (+0.339 eV; -0.341 eV).
6. Page 9, Line 233: 0.05+/-0.01: how this error was obtained?
Answer: The error corresponds to the step width of the electron energy scan (10 meV). We added this information in the revised manuscript, where we mention this error. Since a sharp maximum is obtained for this peak, such error estimation is justified.
7. Page 9, Line 278: “nitroimidazolic anions”: it would be helpful to give the formula
Answer: We added the corresponding formulas in this section 2.1.2.3..
Page 10, Line 299: “… they were extracted by a weak electric field between …”: see point 3.
Answer: See answer to point 3.